# Phelan McDermid Syndrome: Multiple Sclerosis as a Rare but Treatable Cause for Regression—A Case Report

**DOI:** 10.3390/ijms22052311

**Published:** 2021-02-25

**Authors:** Sarah Jesse, Jan Philipp Delling, Michael Schön, Tobias M Boeckers, Albert Ludolph, Makbule Senel

**Affiliations:** 1Department of Neurology, Ulm University, 89081 Ulm, Germany; albert.ludolph@rku.de (A.L.); makbule.senel@uni-ulm.de (M.S.); 2Institute of Anatomy and Cell Biology, Ulm University, 89081 Ulm, Germany; jan.delling@uni-ulm.de (J.P.D.); michael.schoen@uni-ulm.de (M.S.); tobias.boeckers@uni-ulm.de (T.MB.); 3Deutsches Zentrum für Neurodegenerative Erkrankungen (DZNE), 89081 Ulm, Germany

**Keywords:** multiple sclerosis, autoimmune diseases, Phelan McDermid syndrome, genetic autism spectrum, regression, SHANK3

## Abstract

Phelan McDermid syndrome (PMcD) is a neurogenetic disease associated with haploinsufficiency of the *SHANK3* gene due to a spectrum of anomalies in the terminal region of the long arm of chromosome 22. *SHANK3* is the abbreviation for SH3 domain and ankyrin repeat-containing protein, a gene that encodes for proteins of the postsynaptic density (PSD) of excitatory synapses. This PSD is relevant for the induction and plasticity of spine and synapse formation as a basis for learning processes and long-term potentiation. Individuals with PMcD present with intellectual disability, muscular hypotonia, and severely delayed or absent speech. Further neuropsychiatric manifestations cover symptoms of the autism spectrum, epilepsy, bipolar disorders, schizophrenia, and regression. Regression is one of the most feared syndromes by relatives of PMcD patients. Current scientific evidence indicates that the onset of regression is variable and affects language, motor skills, activities of daily living and cognition. In the case of regression, patients normally undergo further diagnostics to exclude treatable reasons such as complex-focal seizures or psychiatric comorbidities. Here, we report, for the first time, the case of a young female who developed progressive symptoms of regression and a dystonic-spastic hemiparesis that could be traced back to a comorbid multiple sclerosis and that improved after treatment with methylprednisolone.

## 1. Case Report

In our special consultation service, we saw a mid-twenties female with genetically proven Phelan McDermid syndrome (PMcD) due to a deletion of 681 kb on chromosome 22q13.33. Consistent with this diagnosis, the patient revealed expressive and receptive speech delay, muscular hypotonia, cognitive decline and symptoms of the autism spectrum like repetitive stimulating behavior, object permanence and impaired eye-contact.

A medical history by proxy illustrated bradycardia in the 24th week of gestation due to aortic coarctation that was corrected by surgery postpartum with no complications and complete reversion of cardiac symptoms.

Further development was characterized by hampered motor skills, that were characterized as gross motor impairment with delayed achievement of childhood milestones like crawling, sitting, walking. Under regular physiotherapy, motor skills improved, leading to the ability to walk at the age of 2.5 years, but fine motor skills remained impaired. Additionally, there was an expressive and receptive speech delay.

Up to adulthood, the patient was able to walk properly and communicate in four-word phrases with complete speech comprehension. She had a very good photographic memory, could wash and dress herself, was able to handle a fork and spoon and showed continence during daytime. She had a job in a sheltered workplace where she sorted small constructional elements.

At her early twenties, behavior changed with phases of melancholy and apathy, alternating with episodes of screaming for hours. Accompanying these symptoms, she lost her autonomy, the capability to communicate by language, neglected her social contacts, and became incontinent.

For further diagnostics, the patient underwent admission to psychiatry, where she was diagnosed with depression and treated by mood-stabilizing agents (escitalopram, mirtazapine, and risperidone). Exclusion of organic reasons using EEG and laboratory diagnostics revealed struma multinodosa so that she underwent hemithyroidectomy, but this did not influence her clinical condition.

During the following months, the parents remarked a slight but progressive hemisyndrome on the right side. According to the fact that no neurology department was consulted at the time, it is not possible to determine exactly when this hemisymptomatic occurred.

After 2 years of progressive aggravation, the family attended our hospital for the first time. Apart from the regressive symptoms, the patient had a spastic dystonia with slight hemiparesis. To exclude brain diseases responsible for the symptoms described, we performed MRI, lumbar puncture and long-term EEG (the latter was normal).

MRI and lumbar puncture were conducted under analgosedation. The spinal imaging showed right lateralized lesions at cervical vertebrae 4, cerebral MRI some white matter lesions in the pontine and left temporal areas, all without contrast enhancement (Figure 1A,B).

The cerebrospinal fluid (CSF) demonstrated normal lactate and total protein, a cell count of five leucocytes/µL and an activated cytopreparation with lymphocytes and plasma cells. We found intrathecal immunoglobulin synthesis of IgG and IgM and positive CSF-specific oligoclonal IgG bands, along with a positive measles, rubella and zoster (MRZ) reaction pointing to a chronic-inflammatory disease (Figure 1C). Serum aquaporin4 and myelin–oligodendrocyte–glycoprotein (MOG) antibodies were negative. Additional laboratory diagnostics excluded a rheumatologic origin of our findings.

Under the suspected diagnosis of a multiple sclerosis, the patient was treated with methylprednisolone intravenously. Due to the psychiatric history, we started with a lower dosage of 250 mg for two days, followed by 500 mg for a further 3 days that was well tolerated. During this therapy, no change of symptoms could be detected.

Several weeks later, we saw the patient in our outpatient clinic to again evaluate therapeutic effects. Neurological examination showed dramatic improvement of the spastic-dystonic hemisyndrome which had virtually disappeared. The parents reported an improvement of her gait that was clearly safer so that she could walk for a longer time; she could now use stairs or get out of a car again without assistance. She began to dress and eat autonomously, like she did before. Additionally, she started to talk again in the 2–4-word phrases she used before regression, and returned to her sheltered workplace, where she began to resume her work.

Additionally, her independence concerning daily activities of life improved but did not yet reach the status before the onset of regression. Based on this clinical course, we discussed a long-term immunomodulation with the parents and suggested an oral medication using dimethyl fumarate, as well as control MRI in the course of the disease.

## 2. Discussion

Regression is a relevant manifestation in PMcD that hampers independence and daily activities of living by affecting language and motor as well as self-help skills and cognitive issues [1,2].

Causes for regressive symptoms represent psychiatric disorders like schizophrenia, depression, and bipolar disorder [3,4] that should be assessed by standard neuropsychological evaluation [5]. Additional triggers for regression include a comorbid a comorbid metachromatic leukodystrophy in cases of impairment of the ARSA gene on chromosome 22 [6] that should be taken into account when the MRI shows typical abnormalities that go along with this metabolic disease.

Concerning epilepsy, there are inconsistent data that reveal prevalent seizures and EEG pathology in cases of regression [2]. In contrast, there are also investigations of prolonged awake and asleep states using video-EEG with various EEG abnormalities but without clinical signs of regression during a follow-up period over one year [7].

Here, we report a patient with PMcD with regressive symptoms and the typical MRI and CSF of multiple sclerosis, with clinical improvement after treatment using intravenous methylprednisolone. Thus, the dissemination criteria for MS were fulfilled [8].

The question arises as to whether the occurrence of PMcD and MS is coincidental or whether there is a pathophysiological commonality. As far as we can tell at present, SHANK3 is not expressed in lymphocytes, but in thymic tissue where it is a constituent in the cell cortex of thymocytes [9]. Therefore, SHANK3 may be involved in the coordination of signal transduction in immune cells. It remains open whether there are common pathophysiological aspects, e.g., IGF-1, which is being tested in clinical trials in PMcD [10] and is supposed to show an effect on myelination in the experimental autoimmune encephalomyelitis (EAE, animal model of MS) through stimulation of regulatory T cells [11,12]. As this is the first case describing a comorbid MS in PMcD, it must be seen if there are further patients to confirm the hypothesis of a pathophysiological connection.

## 3. Conclusions

This case underlines the importance of additional diagnostics in patients with Phelan McDermid syndrome when regressive symptoms occur, that should include standard neuropsychological evaluation, cerebrospinal MRI, lumbar puncture, electroencephalography and in selected cases also genetic analyses.

## Figures and Tables

**Figure 1 ijms-22-02311-f001:**
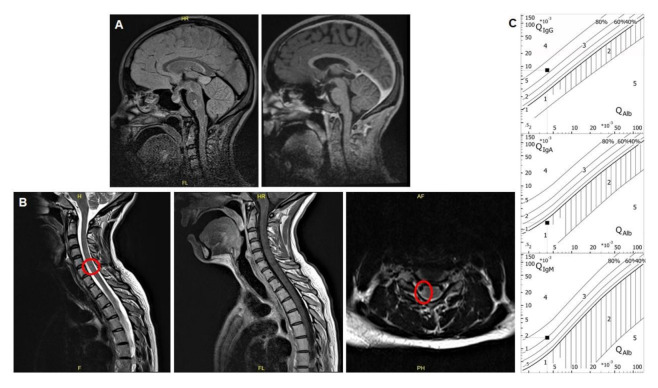
(**A**) T1w sagittal native and after contrast enhancement. Pontine signal hyperintensity without contrast enhancement. A further lesion was detected left temporal (not shown), without contrast enhancement as well. (**B**) T2w sagittal and transversal as well as T1w after contrast enhancement. Right-sided signal hyperintensity without contrast enhancement. (**C**) Results of cerebrospinal fluid investigations depicted in the typical Reibergram with intrathecal immunoglobulin G and M synthesis. 1 = no blood brain barrier dysfunction. 2 = blood brain barrier dysfunction with elevated albumin. 3 = elevated albumin and intrathecal immunoglobulin synthesis. 4 = intrathecal immunoglobulin synthesis only. 5 = incorrect measurement.

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
