# Peer review of "Phelan McDermid Syndrome: Multiple Sclerosis as a Rare but Treatable Cause for Regression—A Case Report"

_ijms, 2021, doi:10.3390/ijms22052311_

Round 1
Reviewer 1 Report
The work by Jesse et al. presents the occurrence Multiple Sclerosis in a patient with Phelan McDermid Syndrome. The study is of great interest for the insight into treatable causes of regression during Phelan McDermid Syndrome and for potential insight into the SHANK3 gene and pathogenetic mechanisms of multiple sclerosis. SHANK3 gene is altered in Phelan McDermid Syndrome. A better introduction of the SHANK3 gene and function would greatly improve a very good Case report.
Additional issues
It should read Phelan McDermid syndrome, not Phelan McDermid syndrom.
Author Response
In the following, we reply to the reviewers' comments on an item-by-item basis:
Reviewer 1
1.) Moderate english changes required.
According to the reviewers’ suggestions, we revised the english style of the manuscript. The changes are marked in the tracking mode.
2.) The work by Jesse et al. presents the occurence of Multiple Sclerosis in a patient with Phelan McDermid syndrome. This study is of great interest for the insight into treatable causes of regression during Phelan McDermid syndrome and for potential insight into the SHANK3 gene and pathogenetic mechanisms in multiple sclerosis. SHANK3 gene is alterend in Phelan McDermid syndrome. A better introduction of the SHANK3 gene and function would greatly improve a very good case report.
We thank reviewer 1 for this annotation and added the following description of the SHANK3 gene and its function in the abstract (page 1, line 10):
„SHANK3 is the abbreviation for SH3 domain and ankyrin repeat-containing protein, a gene that encodes for proteins of the postsynaptic density (PSD) of excitatory synapses. This PSD is relevant for the induction and plasticity of spine- and synapse formation as a basis for learning processes and long-term-potentiation.“
3.) Additional issues: It should read Phelan McDermid syndrome. Not Phelan McDermid syndrom.
This was corrected in the manuscript (page 1, line 2).
Reviewer 2 Report
In this manuscript, the authors describe a fascinating condition in which a PMcD patient developed regression and a comorbid MS. This is a novel, exciting evidence for this comorbidity, which is of high interest.
The manuscript is well written, logically explaining the findings and discuss the results very well.
The following minor comments can improve the manuscript, in my opinion:
- In the case report section, please elaborate in more details the phenotypes described. For example, what are the impaired motor skills exactly? are these fine or gross motor skills? How were they defined?
- What about eye contact? Before and after the regression.
- Are there brain MRI scans of the patient before her early twenties?
- How were the motor capabilities affected as a result of the regression around her early twenties?
- Can the authors further discuss the meaning of normal EEG while some cerebral white matter regions were noticed? The discussion on that matter should be elaborated.
- Figure 1 legend: on second row, "Table 1" is off.
- The clinical phenotypes should be elaborated. "An improvement of her gait" is encouraging, but it will be much more informative if the authors can elaborate on how was this assessed? What did the parents reported: the patient was able to walk for longer time? longer distance? less falls? please elaborate on all clinical features more.
- Why there is no MRI scan image after the treatment? How can the readers learn what the treatment did physiologically? This is a major issue in the manuscript.
- In last row in the discussion "pathophysiologiacal" should be corrected to "pathophysiological"
- Can the authors discuss whether they believe the white matter lesions and MS in PMcD patients can be also responsible to other phenotypes such as intellectual disability and others? Please elaborate why or why not.
- The findings of this interesting case report keep my mind busy as to whether IGF-1 treatment, tested nowadays in clinical trials on PMcD patients, can be relevant also in this case, to block myelination deficits. It can be interesting to discuss this view in the discussion section.
- Thanks!
Author Response
Reviewer 2
In this manuscript, the authors describe a fascinating condition in which a PMcD patient developed regression and a comorbid MS. This is a novel, exciting evidence for this comorbidity, which is of high interest.
We thank the reviewer for the appreciation of our manuscript.
The manuscript is well written, logically explaining the findings and discuss the results very well. The following minor comments can improve the manuscript, in my opinion:
1.) English language and style are fine/minor spell check required.
According also to reviewer 1, we revised the english style of the manuscript. The changes are marked in the tracking mode.
2.) In the case report section, please elaborate in more details the phenotypes described. For example, what are the impaired motor skills exactly? Are these fine or gross motor skills? How were they defined?
Impaired motor skills presented in the early childhood with reduced gross motor skills and delayed achievement of childhood milestones like crawling, sitting, walking. Under physiotherapy, the gross motor impairments improved and remained as fine motor skill impairments.
We changed the following sentence of the manuscript (page 1, line 35):
„Further development was characterized by hampered motor skills, that were characterized as gross motor impairment with delayed achievement of childhood milestones like crawling, sitting, walking. Under regular physiotherapy, motor skills improved, leading to the ability to walk at the age of 2.5 years, but fine motor skills remained.“
3.) What about eye contact? Before and after regression?
There was a good eye contact at the first attending to our hospital without changing after treatment of the regressive symptoms. According to the parents, eye contact was impaired during the crying phase, were a general contact was severely impaired.
With the approval of the reviewer, we would omit this aspect.
4.) Are there brain MRI scans of the patient before her early twenties?
According to the external anamnesis by the parents and the previous findings, no MRI imaging was performed earlier.
5.) How were the motor capabilites affected as a result of the regression around her early twenties?
Due to the regression, initially not a neurologist but a psychiatrist was consulted, so that there are no neurological findings on this. According to the parents, there was little movement during the crying phase and under treatment with mood-stabilizing agents, so that it could not be determined exactly when the hemisymptomatic occurred.
To address this important aspect, we changed the following sentence (page 2, line 13):
„During the following months, the parents remarked a slight but progressive hemisyndrome on the right side. According to the fact that no neurology department was consulted at the time, it is not possible to determine exactly when these hemisymptomatic symptoms occurred.“
6.) Can the authors further discuss the meaning of normal EEG while some cerebral white matter regions were noticed? The discussion on that matter should be elaborated.
It can be assumed that the subcortical localization and only small number of white matter changes do not lead to any tangible changes in the performed surface long-term EEG, such as focus finding.
With the approval of the reviewer, we would omit to discuss this aspect in the manuscript.
7.) Figure 1 legend: on second row, „Table 1“ is off.
We thank the reviewer for this hint and added the missing information (page 2, line 33):
„T1w sagittal native and after contrast enhancement. Pontine signal hyperintensity without contrast enhancement. A further lesion was detected left temporal (not shown), without contrast enhancement as well.”
8.) The clinical phenotypes should be elaborated. „An improvement of her gait“ is encouraging, but it will be much more informative if the authors can elaborate on how was this assessed? What did the parents reported: the patient was able to walk for longer time? Longer distance? Less falls? Please elaborate on all clinical features more.
According to the reviewers suggestion, we elaborated the clinical phenotype after treatment (page 3, line 14):
„The parents reported an improvement of her gait that was clearly safer so that she could walk for a longer time; she could now use stairs or get out of a car again, without assistance. She began to dress and eat autonomously, like she did before. Additionally, she started to talk again in the 2-4 word-phrases she used before regression, and returned to her sheltered workplace, where she began to resume her work.“
9.) Why there is no MRI scan image after the treatment? How can readers learn what the treatment did physiologically? This is a major issue in the manuscript.
This is an important aspect, especially regarding the regressiveness of white matter changes after therapy. Control imaging was recommended to the parents. However, since this would require a new anesthetic, they refrained as the symptoms improved clinically. To address this important aspect, we added the following sentence into the manuscript (page 3, line 25): „Based on this clinical course, we discussed a long-term immunomodulation with the parents and suggested an oral medication using dimethyl fumarate, as well as control MRI in the course of the disease.“
10.) In the last row in the discussion „pathophysiologiacal“ should be corrected to „pathophysiological“
We corrected this in the manuscript (page 3, line 52).
11.) Can the authors discuss whether they believe the white matter leasons and MS in PMcD patients can be also responsible to other phenotypes such as intellectual disability and others? Please elaborate why or why not.
Since this was the first time that white matter lesions were detected in a patient's MRI, one can only speculate that white matter changes, such as are typically found in patients with multiple sclerosis, tend not to explain the clinical phenotype with cognitive, motor and language impairments in Phelan McDermid syndrome.
However, there are initial indications – based on scientific MRI evalutation - that changes in the white matter are also present in this disease (Jesse et al.,2020, Bassell et al., 2020). What relevance this may have for the phenotype or genotype remains to be researched.
Jesse et al. Severe white matter damage in SHANK3 deficiency: a human and translational study. Ann Clin Transl Neurol. 2020, Vol 7, pages 46-58.
Bassell et al. Diffusion Tensor Imaging Abnormalities in the Uncinate Fasciculus and Inferior Longitudinal Fasciculus in Phelan-McDermid Syndrome. Pediatr Neurol. 2020, Vol 106, pages 24-29.
12.) The findings of this interesting case report keep my mind busy as to whether IGF-1 treatment, tested nowadays in clinical trials on PMcD patients, can be relevant also in this case, to block myelination deficits. It can be interesting to discuss this view in the discussion section. Thanks!
This is an interesting aspect! Accoding to this suggestion, we added the following sentences into the discussion section (page 3, line 47) and added the following literature (page 4, line 34):
“It remains open whether there are common pathophysiological aspects, e.g., IGF-1, which is being tested in clinical trials in PMcD (Kolevzon et al) and is supposed to show an effect on myelination in the experimental autoimmune encephalomyelitis (EAE, animal model of MS) through stimulation of regulatory T cells (Bilbao et al, Webster et al.).”
Bilbao D, Luciani L, Johannesson B, Piszczek A, Rosenthal N. Insulin‐like growth factor‐1 stimulates regulatory T cells and suppresses autoimmune disease. EMBO Mol Med. 2014, 6:1423-1435
Webster HD, Growth factors and myelin regeneration in multiple sclerosis. Mult Scler. 1997 Apr;3(2):113-20
Kolevzon A, Bush L, Wang AT, Halpern D, Frank Y, Grodberg D, Rapaport R, Tavassoli T, Chaplin W, Soorya L, Buxbaum JD. A pilot controlled trial of insulin-like growth factor-1 in children with Phelan-McDermid syndrome. Molecular Autism. 2014, 54